# Global Evolutionary Analysis of 11 Gene Families Part of Reactive Oxygen Species (ROS) Gene Network in Four *Eucalyptus* Species

**DOI:** 10.3390/antiox9030257

**Published:** 2020-03-21

**Authors:** Qiang Li, Hélène San Clemente, Yongrui He, Yongyao Fu, Christophe Dunand

**Affiliations:** 1Citrus Research Institute, Southwest University/Chinese Academy of Agricultural Sciences, Chongqing 400712, China; liqiang@cric.cn (Q.L.); heyongrui@cric.cn (Y.H.); 2Laboratoire de Recherche en Sciences Végétales, CNRS, UPS, Université de Toulouse, 31326 Castanet-Tolosan, France; sancle@lrsv.ups-tlse.fr; 3School of advanced agriculture and bioengineering, Yangtze Normal University, Chongqing 408100, China; yongyaofu@yznu.edu.cn

**Keywords:** reactive oxygen species, Eucalyptus, expert annotation, divergence time, peroxidases, evolutionary rate

## Abstract

*Eucalyptus* is a worldwide hard-wood species which increasingly focused on. To adapt to various biotic and abiotic stresses, *Eucalyptus* have evolved complex mechanisms, increasing the cellular concentration of reactive oxygen species (ROS) by numerous ROS controlling enzymes. To better analyse the ROS gene network and discuss the differences between four *Eucalyptus* species, ROS gene network including 11 proteins families (1CysPrx, 2CysPrx, APx, APx-R, CIII Prx, Diox, GPx, Kat, PrxII, PrxQ and Rboh) were annotated and compared in an expert and exhaustive manner from the genomic data available from *E. camaldulensis*, *E. globulus*, *E. grandis,* and *E. gunnii*. In addition, a specific sequencing strategy was performed in order to determine if the missed sequences in at least one organism are the results of gain/loss events or only sequencing gaps. We observed that the automatic annotation applied to multigenic families is the source of miss-annotation. Base on the family size, the 11 families can be categorized into duplicated gene families (CIII Prx, Kat, 1CysPrx, and GPx), which contain a lot of gene duplication events and non-duplicated families (APx, APx-R, Rboh, DiOx, 2CysPrx, PrxII, and PrxQ). The gene family sizes are much larger in *Eucalyptus* than most of other angiosperms due to recent gene duplications, which could give higher adaptability to environmental changes and stresses. The cross-species comparative analysis shows gene gain and loss events during the evolutionary process. The 11 families possess different expression patterns, while in the *Eucalyptus* genus, the ROS families present similar expression patterns. Overall, the comparative analysis might be a good criterion to evaluate the adaptation of different species with different characters, but only if data mining is as exhaustive as possible. It is also a good indicator to explore the evolutionary process.

## 1. Introduction

Due to their fast growth rate, valuable wood and fibre properties, and wide adaptabilities, *Eucalyptus* species with a haploid chromosome number of 11 have been rapidly introduced from Australia to other countries such as France, Brazil, Portugal, and China. Over 700 species constitute the *Eucalyptus* genus which have different growth conditions and phenotypes [1]. In the past 15 years, several studies have led to a better understanding of the *Eucalyptus* genome and the development of an important set of genetic/genomic tools, which can be used to enhance future breeding efforts. Along with the genomic sequencing projects [2] and the expansion of the expressed sequence tag (EST) libraries for some species such as *E. camaldulensis*, *E. globulus*, *E. grandis,* and *E. gunnii*, expert annotation and comparative analysis have become possible. These four species originated from different environments, with different morphologies and genotypes. *E. camadulensis* is a species widely represented throughout Australia, very tolerant to salt stress and drought, and a variable tolerance to cold depending on the origin. The trees are most often of medium size and they are highly cultivated. *E. globulus* is mainly located in south Australia and in Tasmania island, is among the tallest tree in the world. *E. grandis* grows in the east coast and sub-coast. The adult trees are medium to large. *E. gunnii* is an endemic species of Tasmania island, found in mountainous regions and tolerant up to –18 ° C. The adult tree is medium size with many branches and round leaves. *Eucalyptus* species are very reactive after stress, explaining the competitiveness of eucalyptus trees for the occupation of space. 

To adapt to various biotic and abiotic stresses, plants have evolved very complex regulatory mechanisms that can increase the cellular concentration of reactive oxygen species (ROS), including singlet oxygen (^1^O_2_), superoxide anion (O_2_^−^), hydrogen peroxide (H_2_O_2_), and hydroxyl radical (HO^•^). A high level of ROS is highly toxic and will lead to oxidative cell damage. When their concentrations are controlled, they participate in some positive biological processes such as cell growth [3], programmed cell death [4], and signalling [5]. The level of ROS in cells is determined by interplay between ROS producing pathways and scavenging mechanisms. During the evolution processes of plants, efficient ROS scavenging mechanisms have been developed allowing a tight regulation of the ROS homeostasis. Proteins able to regulate the ROS homeostasis (production and scavenging) are part of the ROS gene network [6]. In this study, genes, proteins, and families belonging to this network will be named ROS genes, ROS proteins, and ROS families respectively.

Among this network, peroxidases are enzymes able to reduce H_2_O_2_ by oxidizing various substrates such as lignin subunits, lipid membranes, and some amino acid side chains or regulate the ROS homeostasis [7,8]. These enzymes are present in all kingdoms and play very important roles in plants from germination to senescence, as well as during defence mechanisms, immune responses and pathogeny [8,9,10]. They can be divided into heme and non-heme peroxidases according to the presence or absence of a protoporphyrin IX and Fe (III) complex. In plants, the ROS gene network comprise 10 multigenic families of peroxidases: ascorbate peroxidase (APx), ascorbate peroxidase related (APx-R), catalase (Kat), glutathione peroxidase (GPx), alpha-dioxygenase (DiOx), Class III peroxidases (CIII Prx), and four small and highly conserved families, namely 1-Cysteine peroxiredoxin (1CysPrx), Typical 2-Cysteine peroxiredoxin (2CysPrx), Atypical 2-Cysteine peroxiredoxin type II (PrxII), and type Q (PrxQ) which belong to the peroxiredoxin superfamily [11]. In addition, the plant respiratory burst oxidase homologues (Rboh), also known as NADPH oxidases, belonging to the ROS gene network even if not members of peroxidase families, have been analysed. Present in all kingdoms, these proteins are responsible for the superoxide generation [12]. Genes belonging to multigenic families are often subjected to gene gain or loss events [13]. Gene duplications are a major source of differences between species during the process of evolution and this is more evident when the species are subject to selection pressures and restrictive growth conditions [9]. Therefore, the study of the relationships between heterogeneous data such as genome structure, gene structure, gene gain/loss and function across different species or strains is necessary for the large scale evolution and adaptability analyses [14]. The explosion of sequencing projects resulted in the production of a large amount of data obtained from automatic annotation procedures. However, the quality of the automatic annotation needs to be improved and completed with manual curation in order to obtain a more accurate and global analysis [15]. Finally, a complementary annotation procedure with experimental detection and EST mining would be necessary to overcome the risks associated with assembly and annotation biases [9,15]. The gene conservation and variation should be a crucial criterion to evaluate the capabilities of some organisms to adapt to different environmental variations. However, no direct evidence has yet demonstrated that the gain/loss events of some genes between the four *Eucalyptus* species could be responsible of the morphological and physiological characteristics differences. Comparative analysis of the molecular sequence data is essential for reconstructing the evolutionary history of species and inferring the nature and extent of selective forces shaping the evolution of genes and species [16].

The ratio between non-synonymous (Ka) and synonymous (Ks) nucleotide substitution is an indicator of selective pressures on genes and can be used to identify protein coding genes that may have changed function [17]. A ratio significantly greater than 1 indicates positive selective evolutionary pressures while a ratio less than 1 indicates the negative selective pressures which could conserve sequence with few mutations during the process of evolution [18,19,20]. For several decades, the well-known hypothesis called “molecular clock” has been used. This hypothesis proposes a roughly proportional relationship between the amino acid substitutions and the separation time between compared species. Substitution rate can be used for divergence analysis which is very useful for the evolutionary research [21]. Estimating the divergence time is generally more difficult than reconstructing of a phylogenetic tree, because genes do not evolve at a constant rate. For this reason, authors have used many independently evolving genes to estimate divergence times in the hope of reducing the effect of rate variation [22,23]. In recent years, a large number of authors have investigated the evolutionary relationships of plants using both molecular and paleontological data [24,25]. BEAST (Bayesian evolutionary analysis sampling trees) is a powerful and flexible evolutionary analysis bioinformatics package for molecular sequence variation. It also provides a resource for the further development of new models and statistical methods of evolutionary analysis [26].

Comparative analysis of DNA sequences from multiple species is a powerful approach for identifying coding and functional non-coding sequences, as well as sequences that are unique of a given organism [27]. In this study, 11 families belonging to the ROS gene network have been annotated from genomic sequences available from four *Eucalyptus* species. Then, the distribution and the duplication of the ROS genes were analysed, followed by a global comparative analysis including gene gain and loss, family conservation, and expression study based on publicly available EST libraries and RNA-seq (RNA Sequencing) data. The evolutionary rate and the divergence time among species have also been studied. Comparative genomic analysis can show differences between the genomes of very closely related species. These differences could then shed light on the phenotypic differences between the four species. Genes that differ between species can be studied deeply to determine their roles in phenotypic differences. To our knowledge, this is the first time that a large-scale and expert annotation and comparative analysis have been performed on four *Eucalyptus* species, allowing to analyse the duplication events in the process of evolution and the divergence between different organisms.

## 2. Materials and Methods

### 2.1. Source of Genomic and Protein Sequences

The genome and the proteome of *E. camaldulensis* and *E. grandis* were downloaded respectively from Kazusa Database (http://www.kazusa.or.jp/eucaly/index.html) and Phytozome (http://www.phytozome.net/Eucalyptus.php) a joint project of the Department of Energy’s Joint Genome Institute (JGI) and the Centre for Integrative Genomics (CIG). Sequencing program of *E. globulus* clone X46 and *E. gunnii* clone FCBA #634 genomes (Cagire Azura, protected under UPOV number 20070559) were obtain from JGI and Tree for Joule respectively. The annotated peroxidase sequences from the four *Eucalyptus* species, *Arabidopsis thaliana*, *Vitis vinifera*, *Populus trichocarpa,* and *Medicago truncatula* can be found in the RedoxiBase database (http://peroxibase.toulouse.inra.fr) [28,29].

### 2.2. Data Mining and Expert Annotation

An expert strategy was used for data mining and annotation (Figure 1). To obtain the automatically predicted peroxidases, the protein sequences of *P. trichocarpa* (retrieved from RedxiBase) were used as a query to search the proteome of *E. grandis* and obtain an initial set of automatically annotated proteins corresponding to the 11 families, followed by an expert process to discard prediction errors. In this process, alternative transcript variants and redundant sequences were discarded to prevent artefact during phylogenetic analysis. Partial gene models were verified based on gene structure, presence of conserved domains, and EST supports. The corrected set of protein sequences were used for genome homology prediction using Scipio [30] to obtain the corresponding chromosomal positions, gene structures, DNA, and CDS sequences. New paralogs have been obtained and added to the initial protein set. Each gene has been named as following: Egr, followed by the protein family abbreviation and by a number which represents the position order on the chromosomes. The annotation protocol for *E. camaldulensis* was performed using *E. grandis* sequences previously annotated, following a similar three-step process.

The short-read of *E. globulus* genomic DNA was assembled with mapping method [31]. The reads were assembled using *E. grandis* peroxidase DNA sequences as reference, building a sequence that is similar but not necessarily identical to the base sequence, visualized with software Tablet version 1.12 [32] and manually annotated. A similar strategy has been used for *E. gunnii* annotation using the proteins detected in the three other *Eucalyptus* species as a query set. The sequences of *E. camaldulensis*, *E. globulus*, and *E. gunnii* were named according to the *E. grandis* orthologous sequences.

### 2.3. Pairwise Comparison and Search for Missed Peroxidase Sequences

Because closely related species have similar genomes, the sequenced genes of one species can be used as a template to design primers to search the missed related sequences or to complete partial sequences or to modify the pseudogenes. A gene gain/loss analysis has been performed according to the % of identities between protein sequences of these four species. 

In order to search for putative missed sequences, 90 pairs of PCR primers have been designed (Appendix A) based on the DNA, promoter and terminator sequences available for at least one organism. The genomic DNA of the four *Eucalyptus* species was extracted from leaves with the cetyl-trimethyl-ammonium bromide (CTAB) method. For each pair of primers, three contrasted annealing temperatures (53 °C, 58°C and 63 °C) were used coupled with an elongation temperature of 72 °C and a pre- and denaturation temperature of 94 °C. In the reaction of each pair of primers, at least one positive control has been used with the DNA containing the target gene as the template. The PCR products of the newly found genes were sequenced, assembled, annotated and deposited in RedOxiBase. The annotation process containing automatic annotation, manual annotation, and experiments for ‘missed’ genes can be considered as the comprehensive annotation processes (Figure 1). The Venn diagrams of the ROS genes in the four *Eucalyptus* species constructed by NetVenn [33].

### 2.4. Analysis of Phylogeny, Chromosomal Localization and Duplication Events

The phylogenetic analysis was conducted with all the complete protein sequences of *E. grandis* aligned using MAFFT with default parameters [34] and further inspected and visually adjusted using BioEdit version 7.2 [35]. The phylogenetic trees were reconstructed with the Maximum-likelihood (ML) method using PhyML-aLRT 3.0 [36] and edited with Mega 6 [37]. The graphical presentation of gene localisation of the 11 families on *E. grandis* chromosomes and the duplication linkage between genes were produced using MapChart V2.1 [38]. The duplication events of *E. grandis* including whole genome duplication (WGD), segmental duplication (SD) and tandem duplication (TD) [9] were analysed based on comparative phylogeny between *A. thaliana* and *E. grandis*. 

### 2.5. Expression Analysis Based on ESTs and RNA-Seq Data

To analyse the expression patterns of ROS gene families, data were retrieved from EST libraries and RNA sequencing projects. The expression data of the whole set of annotated proteins have been analysed using an alignment (tBlastN) against the EST libraries of the four *Eucalyptus* species available on NCBI, while the EST data of *A. thaliana* were obtained from The Arabidopsis Information Resource (TAIR) (https://www.arabidopsis.org/). To analyse the relationship between gene duplication and gene expression profiles, the RNA-seq data, available from six tissues (root phloem, immature root xylem, roots, mature leaf, young leaf, shoot tip and flower) of *E. grandis* [2], have been visualised for the ROS genes by heatmap using software Expander V6 [39]. The average EST or RNA numbers of each family in the five organisms were calculated from the EST and RNA-seq data and the expression level of each family in each organism were defined with the following formula: average numbers of ESTs of a family / total EST numbers of ROS gene network.

### 2.6. Analysis of Evolutionary Rate and Divergence Time

The coding DNA sequences (CDS) from the complete sequences were used for the Ka and Ks analysis. The consensus parts of the CDS sequences were used for the analysis with the DNAsp 5.0 software [40]. The tree was produced by Beast v1.8.0 [26] with the chimeric CDS sequences, annotated with TreeAnnotator v1.8.0 and finally visualized with FigTree v1.4.0. 

## 3. Results and Discussion

### 3.1. Data Retrieval, Semi-Automatic Annotation and Statistics

Thanks to the semi-automatic and manual protocol used for the annotation, 884 genes from 11 families have been annotated from the four genomes of *Eucalyptus* species including complete, partial and pseudogenes sequences (Table 1). Here, 229 genes of *E. grandis* including 64 pseudogenes have been identified. Only 92 proteins were correctly predicted by Phytozome using homology-based FgenesH and GenomeScan prediction programs [2], while 40 genes were incorrectly predicted. The remaining 97 sequences, not automatically predicted by Phytozome, have been finally annotated manually from the genomic assembly and the support of EST libraries available on NCBI (Appendix A). In *E. camaldulensis*, 82 of the 214 sequences were correctly predicted by Kazusa combining several gene prediction programs (GeneMark.hmm, GeneScan, NetGene2 and Splicepredictor) [41], while 124 genes were incorrectly predicted, meaning that only 8 not predicted sequences have been manually annotated (Appendix A). In *E. globulus* and *E. gunnii*, since no predictions were available, the annotation was performed using *E. grandis* proteins as a template. 232 and 209 genes were respectively annotated, including 70 and 62 pseudogenes respectively (Appendix A). The qualities and completeness of the four genomic data sets were variable. Indeed, 112, 114, 70 and 78 incomplete genes have been obtained from *E. camaldulensis*, *E. globulus*, *E. grandis* and *E. gunnii* respectively. These incomplete sequences (partial genes and pseudogenes) are related to undetermined nucleotide acids or frame shifts probably due to the sequencing quality, low coverage, and mis-assembly or the genomic pseudogenisation. Among these four sequencing and assembly projects, *E. globulus* showed much better outputs compared with the other three organisms due to the sequencing coverage. The elevated level of incorrect or not predicted genes detected in this study is mostly due to the complexity of multigenic family annotation. 

Although the quality of the annotations of new genomes has been improved thanks to new tools for assembly and annotation, the percentage of incorrect or missed annotations remains high [15]. Results obtained from *E. grandis* and *E. camaldulensis* confirmed the bias of automatic annotation process and particularly in the case of large multigenic families. Some genes are still not or not correctly predicted and annotated. 

### 3.2. Necessary and Effective Detection of Missed Genes

The comparative analysis of the gene sets belonging to ROS gene network found in the four *Eucalyptus* species allows identifying the putative genomic “missed” sequences in one or several organisms (Appendix A). Even if the studied families were subjected to many duplications and gene number variations, the differences detected between the four *Eucalyptus* species appeared to be higher than expected.

In order to determine if the missed sequences are due to gene pseudogenisation, gain/loss events or partial genomic sequencing coverage, 90 pairs of primers were designed to clone and sequence the putative missed sequences. 57 new sequences have been identified (19, 4, 15, and 19 in *E. camaldulensis*, *E. globulus*, *E. grandis*, and *E. gunnii* respectively, Table 1 and Appendix A). Most of the newly found genes are CIII Prxs (82.5%). The percentages of new genes detected by PCR in the total gene number are 8.2%, 1.7%, 6.1%, and 8.3% from *E. camaldulensis*, *E. globulus*, *E. grandis*, and *E. gunnii* respectively. These values can inform on the genome coverage and need to be added to the non-annotated or mis-annotated sequences in order to estimate the quality of the genomic sequencing and assembly.

### 3.3. Phylogeny and Chromosomal Localisation of ROS Genes Network

The exhaustive in silico and experimental mining allowed to draw of a global phylogenetic overview of the 11 gene families in *E. grandis* (Appendix A). Each family is well defined and the superfamily membership respected.

Chromosomal localisation of 218 genes part of the ROS gene network allowed a global distribution analysis. 61 of 218 sequences (28%) were located on chromosome 1 among which there were 56 CIII Prxs (32.2% of 174 CIII Prxs), while only five genes (2.3%) have been found on chromosome 4 (Figure 2, Appendix A). The highest concentration of genes is on chromosome 1 (1.52 per Mb) while the lowest is on chromosome 4 (0.12 per Mb). Most of the genes detected on chromosome 1 are located in clusters such as Prx1–08, Prx09–16, Prx17–25, Prx30–41, and Prx50–55 with intergenic distances shorter than 15 Kb (average intergenic distance calculated for the whole genome) which support the theory of hot spots of duplication events already detected for other superfamilies [9]. This can suggest a hot spot of duplications related with high functional priority.

### 3.4. Gene Gain and Loss Events during the Evolutionary Process

The speciation process was accompanied with the birth of organism specific genes [42]. This process is also detected between close species and could be enhanced in multigenic families. Gene gain and loss events have been found between the four species (Appendix A). For example, *E. globulus* contains two specific CIII Prxs orthologous, 11 missed, and 181 common sequences compared with CIII Prxs family of *E. grandis*. Concerning *E. gunnii*, 16 CIII Prxs were lost and no gene were gained compared to *E. grandis*, 11 are lost and 6 isoforms were gained compared to *E. globulus*, and 14 are lost and 10 genes were gained compared to *E. camaldulensis*.

Based on the chromosomal location and the phylogenetic analysis, part of the missed genes were members of identified clusters or belong to the pseudogene group (missed genes in one organism are all pseudogenes in other organisms) which support recent evolution events (Figure 2, Appendix A) [9]. For example, 1CysPrx03-2, missed in *E. grandis* genome, is a duplication of 1CysPrx03-1 (Table 2). Prx183, Prx189 and GPx11 missed in *E. grandis* are pseudogene in the three other *Eucalyptus* species. In this case, these gene loss events might not have such an effect on plant biology due to the gene redundancy or to the mis-functionality of pseudogenes. In contrast, missed genes which are singletons in other organisms could directly introduce different features for the different *Eucalyptus* species. The loss of functional genes might be the crucial reason why the four organisms possess different biological characteristics and adaptabilities.

Among the CIII Prxs, 155 sequences are common to the four *Eucalyptus* species. One sequence has been found to be specific to *E. grandis* and 3 to *E. camaldulensis*. However, in *E. globulus* and *E. gunnii* no specific CIII Prx genes were found (Figure 3, Appendix A). Regarding the other genetic families, the four *Eucalyptus* species contain similar gene numbers (54, 53, 53, and 53 respectively in *E. camaldulensis*, *E. globulus*, *E. grandis,* and *E. gunnii*). 48 sequences are common to the 4 organisms. No organism specific gene has been detected in the four *Eucalyptus* species. However, it is still difficult to determine if this organism specific gene resulted from a gene gain event or a gene lost event in other organisms. Anyway, the pairwise comparison should be still considered as a useful method for the selection of candidate genes that have potentials of organism specific functions like EguPrx171 with no paralogs in the other three organisms. The number of gene absents from *E. gunnii* (23) is higher than the other species which indicate the possibility that there were fewer ROS genes especially CIII Prx genes. This might be the reason why the ROS gene network size of *E. gunnii* is smaller than other three organisms.

### 3.5. ROS Gene Families Possess Different Features of Conservation

Phylogenetic analysis and the chromosomal localisation, allow identifying various duplication events such as TD, SD, and WGD events. Based on the duplication events, the 11 families have been be categorized into duplicated gene families such as CIII Prx, Kat, 1CysPrx, and GPx and non-duplicated families such as APx, APx-R, Rboh, DiOx, 2CysPrx, PrxII, and PrxQ. 

In order to determine if the observations made for the four *Eucalyptus* species are also valid for more distant species, compositions of the ROS genes network have been analysed for four other dicotyledon organisms: *A. thaliana*, *V. vinifera*, *M. truncatula,* and *P. trichocarpa* (Table 3). Non-duplicated families (APx, APx-R, Rboh, DiOx, 2CysPrx, PrxII, GPx, and PrxQ) present a stable isoform number between the four more distant dicotyledon organisms (Table 3). However, CIII Prx, 1CysPrx and Kat which are families subjected to size variation and duplications between *Eucalyptus* species, are also variable between the selected more distant dicotyledon organisms with a large increase of gene numbers in *Eucalyptus* species compared to the other one. 

Special attention was paid to detect and identify the partial sequences and pseudogenes for global analysis. Indeed, the presence of pseudogenes in some species can reflect recent duplication events that are disappearing. However, the conservation of these duplications can be necessary to adapt to the contrasted and extreme environments. Taking this into account and according to the research of Chen [43], the gene families with conserved size and little pseudogenes such as APx-R, 2CysPrx, DiOx, PrxQ, APx, GPx, Rboh, and PrxII could be essential genes and subjected to less function variations. On the other hand, gene families with large size variation and pseudogenes, such as CIII Prx, 1CysPrx, and Kat, could contain functional redundancy but spatio-temporal specificity necessary for rapid adaptation. For a further research, most genes belonging to families with conserved size should be potential candidates for simple but crucial functions.

### 3.6. Families with Size Variation Contain a Lot of Gene Duplication Events

CIII Prx, 1CysPrx, and Kat families present size variations between close or more distant species. These differences are mainly due to the different levels of duplications. From our recent research 80 TDs, eight SDs, and 10 WGDs have been found for CIII Prx family of *E. grandis*. For example, one large and recent SD can be found on chromosome 1 including EgrPrx01–08 and EgrPrx09–16 (Figure 4). These two regions result from a recent SD with eight genes obtained from seven older and successive TDs (Figure 5A). Genes derived from a duplication event have a very similar coding region from the sequences and the gene structures point of view. One ancestral gene was duplicated seven times and formed the EgrPrx01–08 segment. Then this segment was duplicated and reversely inserted into chromosome 1 to form the EgrPrx09–16 segment (Figure 5B). The paralogs of these 16 genes can be detected in the other three *Eucalyptus* species (except the paralog of EgrPrx16 missed in *E. gunnii*) but not in the other organisms such as *A. thaliana*, *P. trichocarpa* and *M. truncatula*. Hence, the duplication events described above should have occur after the divergence of the *Eucalyptus* genus and before the divergences of the four *Eucalyptus* species. Similarly, recent duplication events have been observed for both Kat and 1CysPrx families. Indeed, 9 TDs (including Kat01-1, 01-2, 02 and Kat03, 04, 05 and 06) and 1 TD (including 1CysPrx01, 02) can also be detected for these two families.

### 3.7. Expression Profiles of ROS Gene Families within and among Species

Expression profiles of members of the 11 gene families within and among species show similar expression levels for highly expressed orthologs such as PrxII03, 2CysPrx01 and APx03 (Appendix A–S5, S8–S10). However, some orthologs can present different expression profiles. For example, Prx113 is only highly expressed in *E. camaldulensis* and *E. globulus*, Kat03 is highly expressed in *E. camaldulensis* but lowly expressed in the other three organisms. Orthologs with different expression profiles might be related to the different properties of the four species. The expression of the 11 ROS gene families shows similar profiles for the four *Eucalyptus* species but different from *A. thaliana*. Between families, the expression levels of 2CysPrx and PrxII are absolutely higher than the other families while the expression of 1CysPrxI, APx-R, DiOx, and Rboh are very low (Figure 6) going against Chen’s conclusion that the families with size conservation have higher level of gene expression [44]. In contrast with the expression levels observed in *Eucalyptus* species, the Kat family shows the highest expression and PrxII family the lowest in *A. thaliana*.

Within the *Eucalyptus* genus, the ROS families present similar expression patterns even after the genus being splitted in different species. It can be explained by the fact that species evolving under the natural selection keep some common properties of *Eucalyptus* genus by having a similar expression level of some genes, while along with a long evolution process after the split of the *Eucalyptus* and *Arabidopsis* genera, expression statuses of ROS network families became more and more different from each other.

### 3.8. Duplicated Genes Possess Different Expression Profiles

Regarding the heat map of ROS gene network in *E. grandis*, some groups or sub-groups can be determined based on the expression profiles (Figure 7). No unique relationship has been found between the expression profiles and the duplicated genes. Genes included in SDs, which resulted from large segments containing the coding and regulatory sequences, might have the same expression profile such as SD EgrPrx67, 68 and SD EgrPrx83, 87–88. In contrast, TDs obtained mainly from duplications of coding region only present different expression profiles. For example, EgrPrx62 and EgrPrx63 are highly expressed in mature leaf and root respectively or EgrPrx118–122, among which only EgrPrx121 can be detected, is highly expressed in roots and immature xylem while others are not expressed. In addition, during the evolution process, the duplicated genes can evolve differentially and obtained specific spatio-temporal expression. Even though the question why redundant duplicated genes are found widely has not been answered, the duplication events giving birth to more genes will accumulate mutations faster than a functional single-copy gene, making it possible for one of the two copies to develop new or different functions over generations.

### 3.9. Divergence Dates of the Four Eucalyptus Species

Expert and exhaustive annotation of the four *Eucalyptus* species and *A. thaliana* allowed generating high-quality sequence batch for divergence time analysis. The known divergence between *A. thaliana* and *Eucalyptus* species (112 million years ago (MYA)) has been used to calibrate the time tree and to get the divergence date between the four *Eucalyptus* species [44]. *E. camaldulensis* diverged firstly 1.27 MYA, following by *E. gunnii* 0.89 MYA, and more recently *E. grandis* and *E. globulus* diverged 0.15 MYA (Figure 8). According to the evolutionary rate (Ka/Ks) most ROS genes evolved under negative selection (Appendix A). Interestingly, the CIII Prxs evolved faster than other families based on the higher evolutionary rate. The faster evolution and the high duplication rate could be correlated and due to the intrinsic properties of the CIII Prx family.

## 4. Discussion

In this study, we performed exhaustive and expert annotation and compared 11 gene families part of ROS genes network in four *Eucalyptus* species with a useful and necessary complementary process. This complementary annotation process included manual annotation and correction, and confirmation with EST libraries checks and PCR detection. This provided more expert and exhaustive data in order to reduce the errors produced by automatic assembly and annotation and to obtain the more complete and accurate results. Of course, this method was easier to apply for annotations of closely-related genomes and cannot be applied with far-related ones. Even if the coverage of genomic sequencing is high (more than 90%), we have demonstrated that the bias of automatic annotation is still elevated with over 60% of sequences not or not correctly predicted. Then, the conclusions made from automatic annotation are partial and erroneous. The 11 families studied presented different features of conservation, duplication and expression. During the process of evolution, gene gain and loss occurred between the species. The number of *Eucalyptus* genes belonging to the ROS gene network is much larger compared with some other angiosperms mainly due to recent gene duplications in CIII Prx family during the evolution. 

This large number of ROS genes copies can be easily linked to the *Eucalyptus* species particularities such as its very rapid growth, the persistence of its foliage throughout the year, its relative resistance to drought and to temperatures below freezing. Then this large battery of proteins allows a rapid response to biotic and abiotic stresses and the micro differences observed between the four *Eucalyptus* species may be associated to their environment specificities. 

Indeed, the analysis of ROS genes between *Eucalytpus species* demonstrated recent gain and lost events which confirm that *Eucalytpus* species genomes are very dynamic, as previously demonstrated. These analyses also provided a method for the selection of candidate gene for next functional research. The families of ROS genes possess different expression levels between the different organisms, but similar profiles among *Eucalytpus* species. The explosion and the conservation of the ROS encoding genes numbers may be associated with organ diversification, climatic changes and the constant appearance of new pathogens. Nevertheless, there is still question about the family-explosion: why are some families subjected to numerous duplication events while other protein families have kept a similar gene number after speciation?

Large set of complete sequences collected from the four *Eucalyptus* species allowed determining the divergence time. They diverged from 0.15 MYA to 1.27 MYA. After the divergence of *Eucalyptus* species, most sequences have been maintained by negative selection in the evolutionary process despite speciation, to conserve the main characters of the *Eucalyptus* genus. Our results will help better understand the genetic differences between closely related species, and stimulate additional studies on the mechanisms that underlie speciation and biodiversification.

## Figures and Tables

**Figure 1 antioxidants-09-00257-f001:**
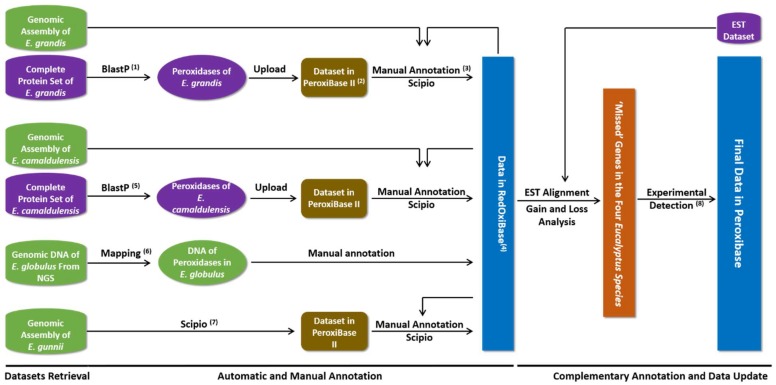
Workflow of the comprehensive annotation process of *Eucalyptus* ROS genes in four *Eucalyptus* species. The comprehensive annotation, starting from the acquisition of data to the storage in a database, consists of automatic annotation, manual annotation and experimental detection as a complimentary annotation. The experimental detection for the missed sequences would only be possible when the relationship between target organisms is very close. (**1**) The query data for this BlastP is the protein sequence set of *P. trichocarpa*. (**2**) The ReroxiBase II is an assistant database of RedOxiBase, which is used for keeping some private temporary data during the annotation process. (**3**) The Scipio program takes the manually annotated protein sequences as the query. (**4**) The data stored in RedOxiBase contain protein, DNA, CDS sequences and the gene structure information for most of the records. Chromosomal positions are also included for the *E. grandis* genes. (**5**) The query for the BlastP similarity search is the protein set of *E. grandis*. (**6**) Mapping program takes the DNA set of *E. grandis* and *E. camaldulensis* as the query to obtain the peroxidase DNA sequences of *E. globulus*. (**7**) The protein set containing peroxidases of *E. grandis*, *E. camaldulensis* and *E. globulus* is used as the query for the Scipio program to obtain the peroxidase data. (**8**) PCR was performed for the “missing” genes among the organisms.

**Figure 2 antioxidants-09-00257-f002:**
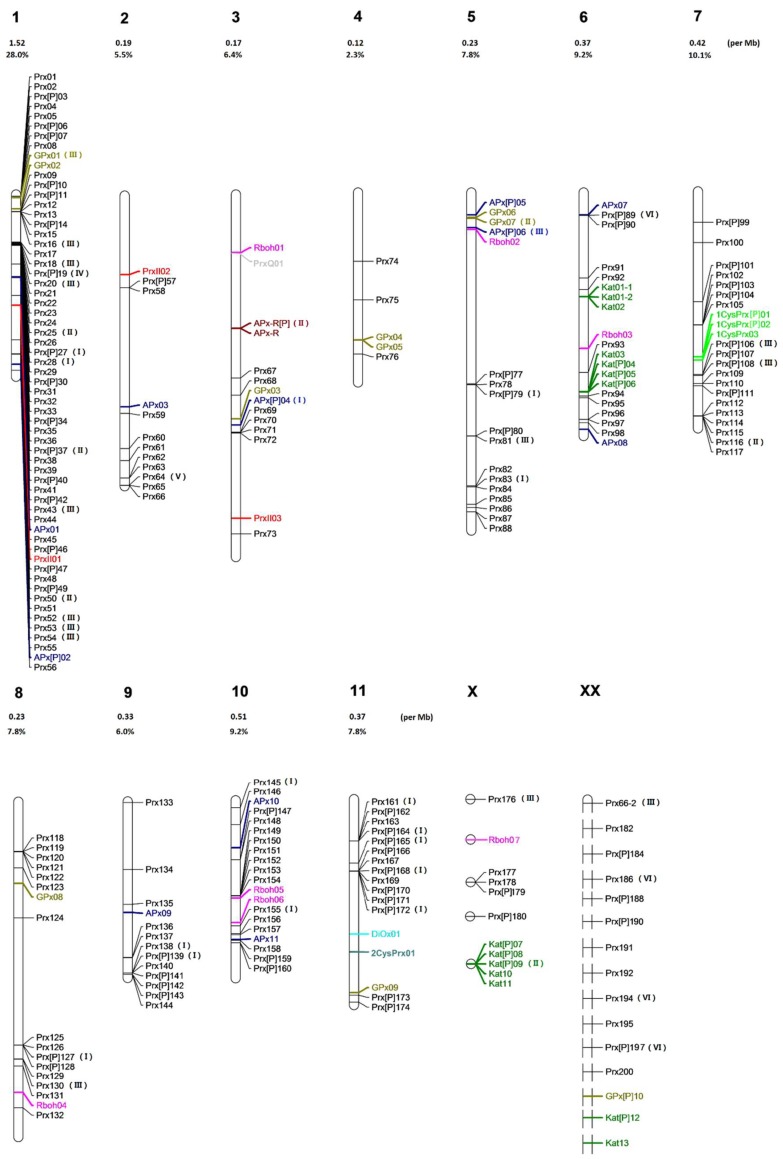
Genomic localization of the ROS gene families from *E. grandis*. All the predicted ROS genes annotated from the genomic program including complete sequences, partial sequences and pseudogenes which can be located on the 11 chromosomes are presented. This synthetic chromosomal localization was displayed by MapChart 2.1. New sequences obtained from cloning strategies are not localized. Different colors represent different families. (I) *E. grandis* specific genes compared to *E. camaldulensis*. (II) *E. grandis* specific genes compared to *E. globulus*. (III) *E. grandis* specific genes compared to *E. gunnii*. (IV) *E. grandis* specific genes compared to *E. camaldulensis*, *E. globulus* and *E. gunnii*. (V) *E. grandis* specific genes compared to *E. camaldulensis* and *E. gunnii*. (VI) *E. grandis* specific genes compared to *E. globulus* and *E. gunnii*. The 11 chromosomes are labelled with 1–11 above each chart. The genes on small scaffolds are visualized on ‘chromosome’ X. The newly found genes without position information are visualized on ‘chromosome’ XX. The ROS gene concentration (ROS gene number per Mb chromosome) and the % of ROS genes on each chromosome (ROS gene number on the chromosome / total ROS gene number in *E. grandis*) are written above the chart without considering the genes on ‘chromosome’ X and ‘chromosome’ XX. The concentration is calculated as formula: gene number / size of the chromosome.

**Figure 3 antioxidants-09-00257-f003:**
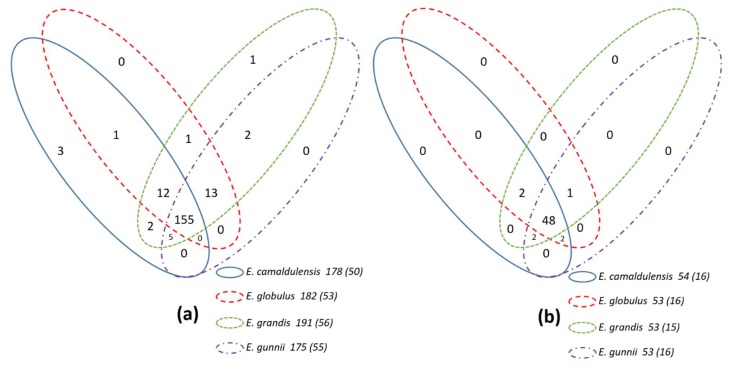
Venn diagrams of the ROS genes in the four *Eucalyptus* species. The Venn diagrams show the numbers of peroxidases gene shared between the four *Eucalyptus* species: *E. gunnii*, *E. camaldulensis*, *E. grandis* and *E. globulus*. (**a**) Venn diagram of CIII Prx family; (**b**) Venn diagram of other 10 ROS gene families. The total gene numbers of each organism, represented by the ovals, were written on the left of each specific name with the numbers of pseudogenes enclosed in brackets. The area of every intersection region and the gene number are not to scale.

**Figure 4 antioxidants-09-00257-f004:**
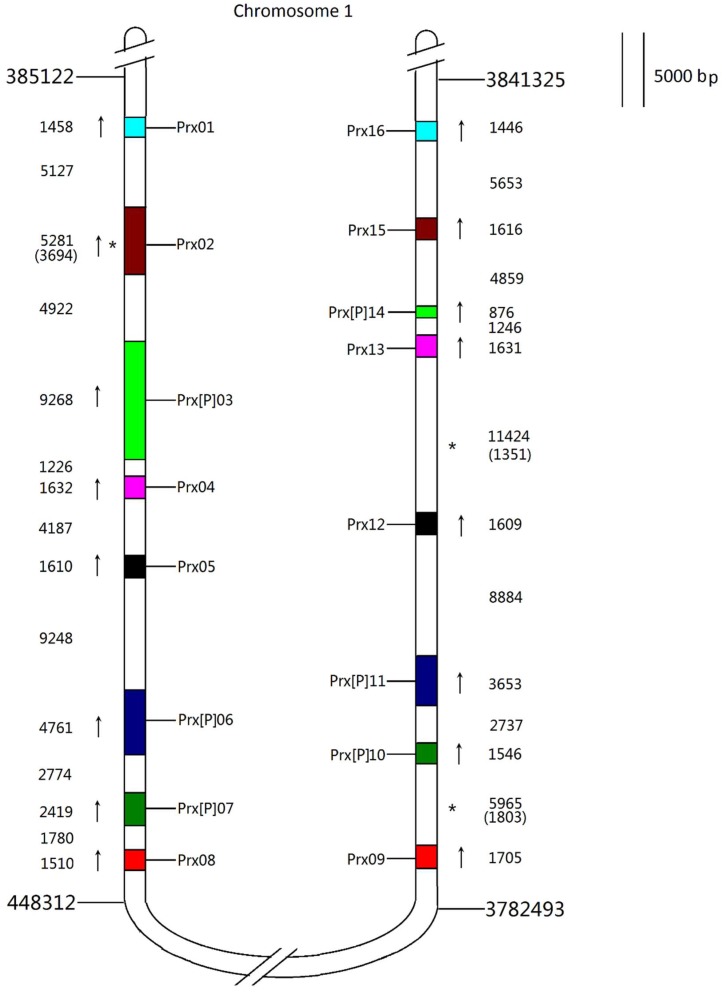
A large duplicated CIII Prx gene cluster on chromosome 1. The sizes of genes and segments are to scale. The start and stop positions of the duplicated segments were written on the top and the bottom of the segments. The colorful regions represent the CIII Prxs in the segments on chromosome 1 and the gene orientations were shown with the arrows on the left or right. The sizes of genes and intervals are noted on the side of each sequence with the numbers of undetermined nucleotide acids in brackets. The homologous genes were displayed with the same color on the two duplicated segments. [P]: the gene has been annotated as pseudogene. *: there are undetermined nucleotide acids (NNN…) in the DNA sequences. The visualization of the chromosomal localization was built by Mapchart 2.1. Arrows represent the gene orientations on chromosomes.

**Figure 5 antioxidants-09-00257-f005:**
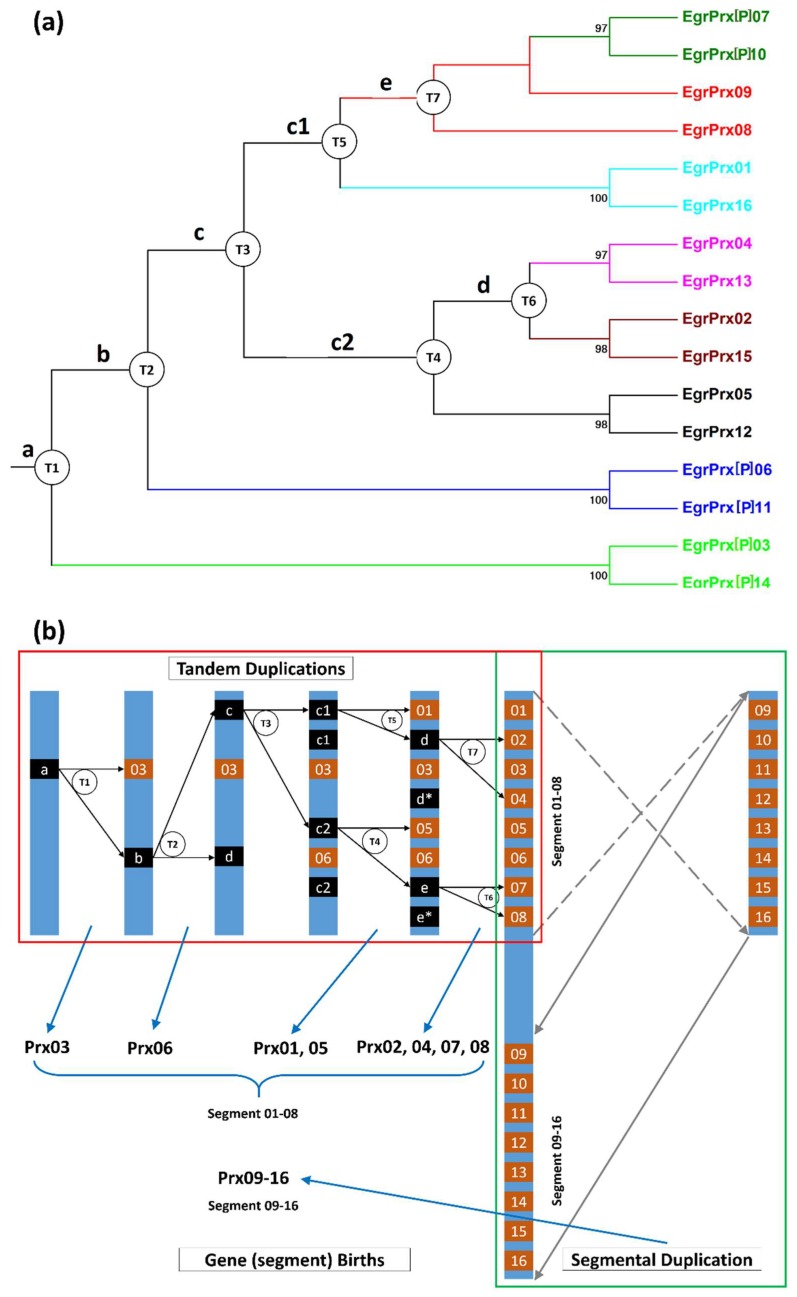
Phylogenetic relationships and hypothetical evolutionary histories of two duplicated segments. **(a)** Phylogenetic tree of the 16 genes. Genes were given different colors according to the colors in Figure 5. The percentage of trees in which the associated taxa clustered together is shown next to the branches. The alignment was performed by MAFFT and the tree was visualized by Mega 6. Cycled letters T with following numbers represent the TD events; **(b)** Schematic diagram of the evolutionary histories of the 16 genes. The putative TD process was visualized in the red rectangle while the SD process in green rectangle. Genes shown with black rectangles (named a, b, c, d and e) represent the ancestors of some CIII Prx genes appeared in the evolutionary process. Asterisk (*) represents the alternative localization of the ancestral gene named with the same letter. T1–T7 represent the tandem duplication events. 1–16 represent Prx01–16. The dotted arrow (grey) in B shows the reversing process of gene segment while the grey arrow represents the genomic insertion event. Cycled letters T with following numbers represent the TD events. The order of gene apparition (genes Prx01–16) was displayed on the lower left corner. Gene sizes are not up to scale.

**Figure 6 antioxidants-09-00257-f006:**
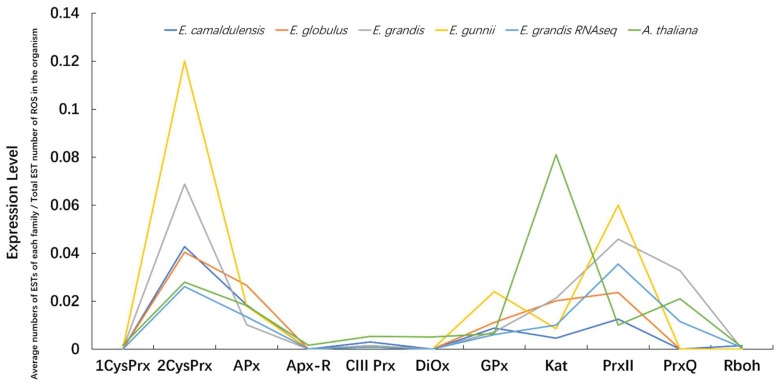
Expression levels of ROS gene families in the four *Eucalyptus* species and *A. thaliana*. Expression levels were calculated by the formula: average EST (or RNA seq reads) numbers of each family / Total EST (or RNA seq reads) number of genes in the organism. EST data were obtained from EST libraries of NCBI.

**Figure 7 antioxidants-09-00257-f007:**
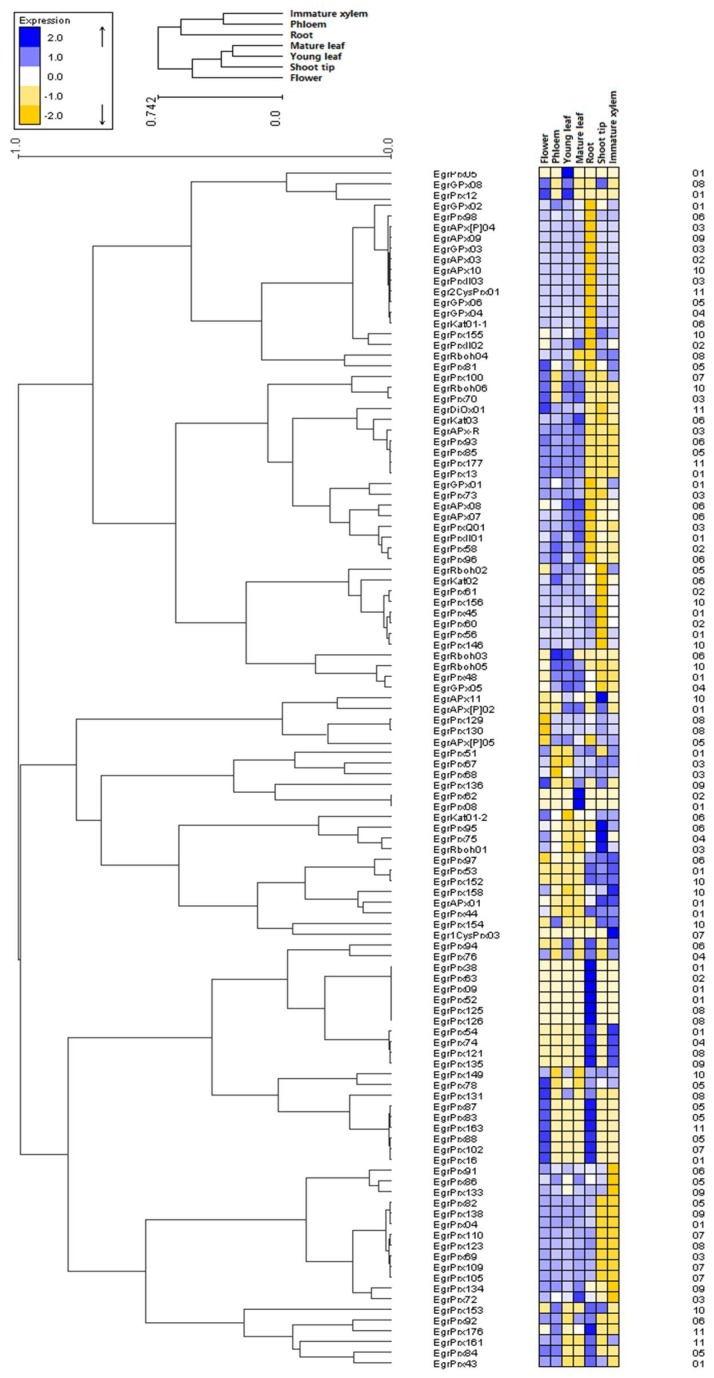
Heatmap of the expression of the ROS genes from *E. grandis* in different tissues. Each line represents a gene and each column represents a tissue. Mature leaf, young leaf, shoot tip, phloem, immature xylem, flower and root have been sampled and tested. The chromosome of each gene is written on the right.

**Figure 8 antioxidants-09-00257-f008:**
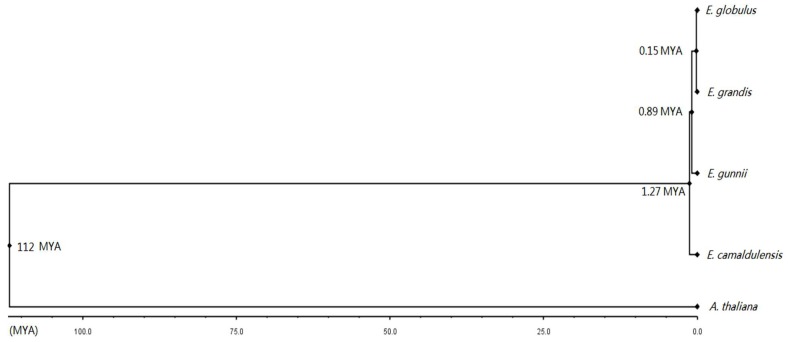
Divergence time of the four *Eucalyptus* species. For each organism the chimerical gene of 49 ROS genes was used in this study. The divergence time between *Eucalyptus* species was written beside the nodes. Million years ago (MYA).

**Table 1 antioxidants-09-00257-t001:** ROS gene numbers of four *Eucalyptus* species annotated from various database sources and from experimental detection performed in this study.

Organisms	*E. camaldulensis*		*E. globulus*		*E. grandis*		*E. gunnii*	
Data Sources	From Databases*	From PCR	From Databases	From PCR	From Databases	From PCR	From Databases	From PCR
1CysPrx	3 (2+0+1)	1 (0+0+1)	4 (0+2+2)	0	3 (1+0+2)	0	3 (1+0+2)	1 (0+0+1)
2CysPrx	1 (0+1+0)	0	1 (1+0+0)	0	1 (1+0+0)	0	1 (1+0+0)	0
APx	10 (3+5+2)	0	10 (7+0+3)	1 (0+0+1)	11 (7+0+4)	0	10 (7+0+3)	0
Apx-R	2 (1+0+1)	0	1 (1+0+0)	0	2 (1+0+1)	0	2 (1+0+1)	0
CIII Prx	163 (84+39+40)	16 (0+6+10)	180 (93+32+55)	3 (0+2+1)	179 (126+2+51)	12 (2+5+5)	159 (100+11+48)	17 (1+8+8)
DiOx	1 (1+0+0)	0	1 (1+0+0)	0	1 (1+0+0)	0	1 (1+0+0)	0
GPx	11 (3+5+3)	0	10 (5+3+2)	0	9 (9+0+0)	1 (0+0+1)	9 (7+1+1)	1 (0+0+1)
Kat	12 (1+4+7)	2 (0+1+1)	14 (3+3+8)	0	12 (2+4+6)	2 (0+1+1)	13 (4+2+7)	1 (0+1+0)
PrxII	3 (3+0+0)	0	3 (2+1+0)	0	3 (3+0+0)	0	3 (2+1+0)	0
PrxQ	1 (1+0+0)	0	1 (0+1+0)	0	1 (1+0+0)	0	1 (1+0+0)	0
Rboh	7 (3+4+0)	0	7 (5+2+0)	0	7 (7+0+0)	0	7 (6+1+0)	0
Total	214 (102+58+54)	19 (0+7+12)	232 (118+44+70)	4 (0+2+2)	229 (159+6+64)	15 (2+6+7)	209 (131+16+62)	19 (1+9+9)
Automatic correct prediction	82 (35.19%)		na		92 (37.70%)		na	
Coverage of Genomic Data	91.8%		98.3%		93.9%		91.7%	

*: The data from the four *Eucalyptus* species were obtained after the annotation of available genomic and EST data. The detail of total number found per organism and per family is written in bracket: including the numbers of complete, partial sequences and theoretical translation or pseudogenes detected separated by plus symbol. The coverage of genomic data corresponds to the following formula:.Genomic coverage = Number of genes from database / (Total number of genes from database + PCR detection). na: no automatic prediction available.

**Table 2 antioxidants-09-00257-t002:** Missed sequences from the genomes of the four *Eucalyptus* species.

Types of Missed Genes	*E. camaldulensis*	*E. globulus*	*E. grandis*	*E. gunnii*
Missed genes in clusters ^1^	Prx19, Prx37, Prx64, Prx79, Prx83, Prx116, Prx127, Prx138, Prx139, Prx161, Prx164	Apx-R[P], Prx19, Prx23, Prx37, Prx39, Prx50, Prx116, Prx129-2, GPx07	1CysPrx03-2, Prx129-2, Prx188	Prx16, Prx18, Prx19, Prx20, Prx52, Prx53, Prx54, Prx64, Prx66-2, Prx106, Prx129-2, Prx130, Prx176, Prx188, GPx01
Singletons ^2^	APx04, Prx27, Prx28, Prx145, Prx155, Prx165, Prx168	Prx89, Prx189, Prx194, Prx195, Prx197, Prx198	Prx183, Prx189, Prx198, GPx11	APx06, Prx81, Prx89, Prx108, Prx183, Prx189, Prx194

^1^ Genes missed in one organism are members of clusters in other organisms; ^2^ Genes missed in one organism are singletons in other organisms; ^Pink^ Genes missed in one organism are all pseudogenes in other organisms; ^Green^ Genes missed in one organism are not pseudogenes in all other organisms. Other missed genes without colour are not all pseudogene and not all partial or complete in other organisms.

**Table 3 antioxidants-09-00257-t003:** Isoform numbers found in 8 organisms.

Multigenic Families	*A. thaliana*	*E. camaldulensis*	*E. globulus*	*E. grandis*	*E. gunnii*	*M. truncatula*	*P. trichocarpa*	*V. vinifera*
1CysPrx	1 (0)	4 (2)	4 (2)	3 (2)	4 (3)	1 (0)	1 (0)	2 (1)
2CysPrx	2 (0)	1 (0)	1 (0)	1 (0)	1 (0)	2 (0)	2 (0)	1 (0)
APx	8 (1)	10 (2)	11 (4)	11 (4)	10 (3)	8 (1)	10 (1)	9 (2)
APx-R	1 (0)	2 (1)	1 (0)	2 (1)	2 (1)	1 (0)	1 (0)	1 (0)
CIII Prx	75 (2)	179 (50)	183 (56)	191 (56)	176 (56)	106 (8)	101 (12)	97 (10)
DiOx	2 (0)	1 (0)	1 (0)	1 (0)	1 (0)	2 (0)	2 (0)	3 (0)
GPx	8 (0)	11 (3)	10 (2)	10 (1)	10 (2)	7 (0)	8 (2)	5 (0)
Kat	3 (0)	14 (8)	14 (8)	14 (7)	14 (7)	1 (0)	4 (1)	2 (0)
PrxII	6 (1)	3 (0)	3 (0)	3 (0)	3 (0)	4 (0)	5 (1)	4 (0)
PrxQ	1 (0)	1 (0)	1 (0)	1 (0)	1 (0)	1 (0)	2 (0)	1 (0)
Rboh	10 (0)	7 (0)	7 (0)	7 (0)	7 (0)	10 (0)	10 (0)	9 (0)
Total	117 (4)	233 (66)	236 (72)	244 (71)	228 (71)	143 (9)	146 (17)	134 (13)

The data from the four *Eucalyptus* species were obtained from the annotation and the PCR detection performed in this study while the data of *A. thaliana*, *V. vinifera* (Grape), *P. trichocarpa* and *M. truncatula* were directly retrieved from RedOxiBase. The number of theoretical translation or pseudogenes is notified in brackets. Each value consists of genes from genomic data, EST data, experimental detection and other sources. Based on the sufficient quantity of genomic and EST data, the values of *A. thaliana* and *M. truncatula* should be able to represent the actual gene numbers even though no experimental detection by PCR were made in this study.

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
