# Peer review of "Global Evolutionary Analysis of 11 Gene Families Part of Reactive Oxygen Species (ROS) Gene Network in Four Eucalyptus Species"

_antioxidants, 2020, doi:10.3390/antiox9030257_

Round 1

Reviewer 1 Report

The manuscript written by Li et al. analyzed the evolutionary process of the ROS gene families in four Eucalyptus species and they discussed the mechanisms how one of the gene family members, CIIIPrx, is diversified in this species. The manuscript is logical and well written. Some minor mistakes were only found and those need to be corrected before publication.

  1. Figure 7: I could not figure out which column represents each tissue expression.
  2. Figure 7: The plant materials used for this analysis would need to be more described: tree age, size and growth conditions, etc., because the expression levels might change depending on such factors.
  3. Line 29: “…duplications.”. The comma should be after this word, not the period.
  4. Line 52: The “PCD” seems to appear just once in the text. So, it does not have to be abbreviated.
  5. Line 128: “Vitis” is correct, not “Vinis”.
  6. Line 181: “missing” should be correct, not “misseing”.
  7. Prx01-08, etc. in the whole text. The hyphen needs to be replaced with the en-dash “–“.

Author Response

  1. Figure 7: I could not figure out which column represents each tissue expression.

The missing information was added in Figure 7.

  1. Figure 7: The plant materials used for this analysis would need to be more described: tree age, size and growth conditions, etc., because the expression levels might change depending on such factors.

The RNA seq data have been generated in the frame of the genome project Myburg et al . The reference has been added and the available information have been included. Unfortunetly, the web page coantaining the RNA seq is not maintained.

  1. Line 29: “…duplications.”. The comma should be after this word, not the period.

Modified.

  1. Line 52: The “PCD” seems to appear just once in the text. So, it does not have to be abbreviated.

Modified.

  1. Line 128: “Vitis” is correct, not “Vinis”.

Modified.

  1. Line 181: “missing” should be correct, not “misseing”.

Modified.

  1. Prx01-08, etc. in the whole text. The hyphen needs to be replaced with the en-dash “–“.

Modified throughout.

Reviewer 2 Report

Global Evolutionary Analysis of 11 Gene Families 3 Part of Reactive Oxygen Species (ROS) Gene 4 Network in Four Eucalyptus species

This study compared 11 gene families part of ROS gene network in four Eucalypatus species with a complementary annotation process. The studies 11 families were presented to have different features of conservation, duplication and expression. Although the study seems important in understanding evolution of the four Eucalyptus species various sections of the manuscript should be revised before manuscript can be accepted.

Comments

Abstract should be rewritten- Study objectives/rationale should be exclusively mentioned. Please shed more light in the important findings.

Line 45: Mention what are the differences in morphology and genotypes in the four Ecucalyptus  species. What different stress conditions are they adapted to?

Why are you particularly looking at these four species only?

Method section should be revised as there are various results in the results sections such as heatmap of the expression of the ROS.. which has not been mentioned before.

Line 15-16: Re-write line 15-16Line 23-24:  Eucalyptus camaldulensis, Eucalyptus 23 globulus, Eucalyptus grandis and Eucalyptus gunnii no need to write full genus name when you mention it for the second time

Line 28, 470 and 473:  eucalyptus- capitalize e and italicize

Line 29- sentence is incomplete -rewrite

Line 181: “missing” instead of “missing”

 Line 345: Taking this into “account”

Line 475, 479: don’t italicize species

Figures: Please replace figure with more clear figures

Author Response

  1. Abstract should be rewritten- Study objectives/rationale should be exclusively mentioned. Please shed more light in the important findings.

The abstract was improved and rewritten.

  1. Line 45: Mention what are the differences in morphology and genotypes in the four Ecucalyptus species. What different stress conditions are they adapted to?

A longer description of the four species have been included in the introduction.

  1. Why are you particularly looking at these four species only?

The availability of both plant materials for DNA extraction and PCR and genomic data were the major criteria to selected these four species. This choice was also supported by the diversity of the Eucalyptus species selected. In addition, perform genetic and phylogenetic analysis with more species was not doable.

  1. Method section should be revised as there are various results in the results sections such as heatmap of the expression of the ROS.. which has not been mentioned before.

Necessary information (about venn diagram and heatmap) was added in section 2.5.

  1. Line 15-16: Re-write line 15-16Line 23-24: Eucalyptus camaldulensis, Eucalyptus 23 globulus, Eucalyptus grandis and Eucalyptus gunnii – no need to write full genus name when you mention it for the second time

Modified.

  1. Line 28, 470 and 473: eucalyptus- capitalize e and italicize

Modified.

  1. Line 29- sentence is incomplete -rewrite

Modified.

  1. Line 181: “missing” instead of “missing”

Modified.

  1. Line 345: Taking this into “account”

Modified.

  1. Line 475, 479: don’t italicize species

Modified.

  1. Figures: Please replace figure with more clear figures

The figures are still in the manuscript but the high-quality figures were uploaded separately.

In addition to the above changes, the authors also proofread throughout to improve the MS. The detailed modification traces were not listed in this letter, please check the file ‘Revised Manuscript – Marked Up’ for details.